# STDP and the distribution of preferred phases in the whisker system

**Nimrod Sherf**[1,2]*, **Maoz Shamir**[1,2,3]

**1** Physics Department, Ben-Gurion University of the Negev, Beer-Sheva, Israel, **2** Zlotowski Center for Neuroscience, Ben-Gurion University of the Negev, Beer-Sheva, Israel, **3** Department of Physiology and Cell Biology Faculty of Health Sciences, Ben-Gurion University of the Negev, Beer-Sheva, Israel

* sherfnim@post.bgu.ac.il

## Abstract

Rats and mice use their whiskers to probe the environment. By rhythmically swiping their whiskers back and forth they can detect the existence of an object, locate it, and identify its texture. Localization can be accomplished by inferring the whisker's position. Rhythmic neurons that track the phase of the whisking cycle encode information about the azimuthal location of the whisker. These neurons are characterized by preferred phases of firing that are narrowly distributed. Consequently, pooling the rhythmic signal from several upstream neurons is expected to result in a much narrower distribution of preferred phases in the downstream population, which however has not been observed empirically. Here, we show how spike timing dependent plasticity (STDP) can provide a solution to this conundrum. We investigated the effect of STDP on the utility of a neural population to transmit rhythmic information downstream using the framework of a modeling study. We found that under a wide range of parameters, STDP facilitated the transfer of rhythmic information despite the fact that all the synaptic weights remained dynamic. As a result, the preferred phase of the downstream neuron was not fixed, but rather drifted in time at a drift velocity that depended on the preferred phase, thus inducing a distribution of preferred phases. We further analyzed how the STDP rule governs the distribution of preferred phases in the downstream population. This link between the STDP rule and the distribution of preferred phases constitutes a natural test for our theory.

## Author summary

The distribution of preferred phases of whisking neurons in the somatosensory system of rats and mice presents a conundrum: a simple pooling model predicts a distribution that is an order of magnitude narrower than what is observed empirically. Here, we suggest that this non-trivial distribution may result from activity-dependent plasticity in the form of spike timing dependent plasticity (STDP). We show that under STDP, the synaptic weights do not converge to a fixed value, but rather remain dynamic. As a result, the preferred phases of the whisking neurons vary in time, hence inducing a non-trivial distribution of preferred phases, which is governed by the STDP rule. Our results imply that the considerable synaptic volatility which has long been viewed as a difficulty that needs to be

**Data Availability Statement:** All relevant data are within the manuscript and its Supporting Information files.

**Funding:** Maoz Shamir received the following grant: The Israel Science Foundation (ISF) grant number 300/16. The funders had no role in study

design, data collection and analysis, decision to publish, or preparation of the manuscript.

**Competing interests:** The authors have declared that no competing interests exist.

overcome, may actually be an underlying principle of the organization of the central nervous system.

## Introduction

The whisker system is used by rats and mice to actively gather information about their proximal environment [1–4]. Information about whisker position, touch events, and texture is relayed downstream the somatosensory system via several tracks; in particular, the lemniscal pathway that relays information about both whisking and touch [5–11].

During whisking, the animal moves its vibrissae back and forth in a rhythmic manner, Fig 1a–1c. Neurons that track the azimuthal position of the whisker by firing in a preferential manner to the phase of the whisking cycle are termed whisking neurons. Whisking neurons in the ventral posteromedial nucleus (VPM) of the thalamus as well as inhibitory whisking neurons in layer 4 of the barrel cortex are characterized by a preferred phase at which they fire with the highest rate during the whisking cycle [1, 6, 12–16], Fig 1d and 1e. The distribution of preferred phases is non-uniform and can be approximated by the circular normal (Von-Mises) distribution

$$\Pr(\phi) = \frac{e^{\kappa \cos(\phi - \psi)}}{2\pi I_0(\kappa)} \tag{1}$$

where $\psi$ is the mean phase and $I_0(\kappa)$ is the modified Bessel function of order 0, Fig 2a. The parameter $\kappa$ quantifies the width of the distribution; $\kappa = 0$ yields a uniform (flat) distribution, whereas in the limit of $\kappa \to \infty$ the distribution converges to a delta function. Typical values for $\kappa$ in the thalamus and for layer 4 inhibitory (L4I) whisking neurons are $\kappa_{\mathrm{VPM}} \approx \kappa_{\mathrm{L4I}} \approx 1$ where $\psi_{\mathrm{VPM}} \approx 5\pi/6$ [rad] and $\kappa_{\mathrm{L4I}} \approx 0.5$ [rad] [2, 13].

Throughout this work we ignore the possible contribution of recurrent connections to the shaping of phase selectivity in layer 4 (but see Discussion). Assuming the rhythmic input to L4I neurons originates solely from the VPM, the distribution of preferred phases of L4I neurons is determined by the distribution in the thalamus and the profile of thalamo-cortical

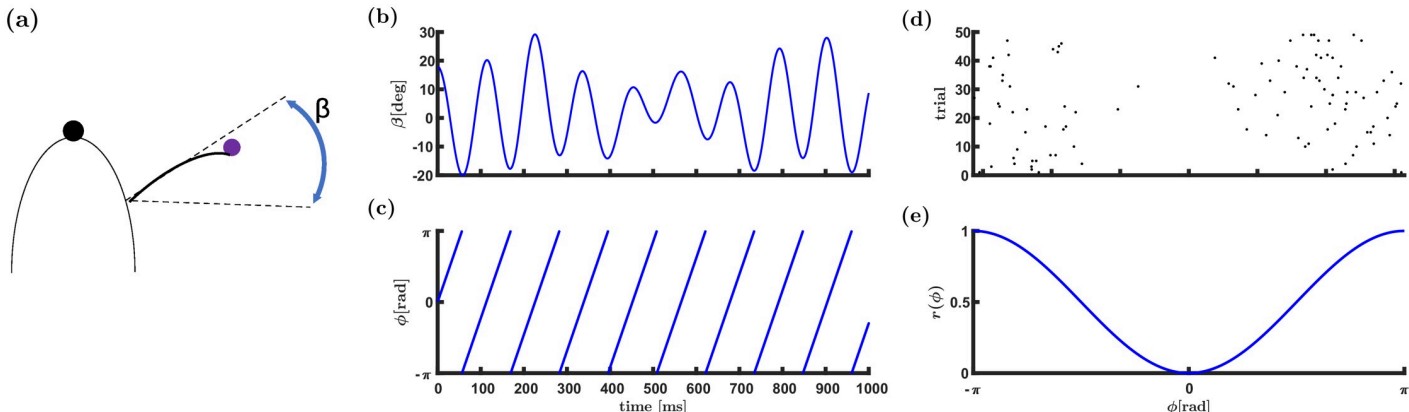

**Fig 1. Representation of the whisking phase.** (a) Mice and rats can infer the azimuthal location of an object by touch. (b) The angular position of a whisker, $\beta$, (during whisking) is shown as a function of time. The angle is often modeled as $\beta(t) = \beta_{\mathrm{midpoint}}(t) + \Delta\beta(t)\cos(\phi(t))$, where $\beta_{\mathrm{midpoint}}(t)$ and $\Delta\beta(t)$ are the midpoint and the whisking amplitude, respectively. (c) The whisking phase $\phi$ as a function of time is $\phi(t) = (\nu t)_{\mathrm{mod}2\pi}$, where $\nu$ is the angular frequency of the whisking. (d) & (e) Raster plot and normalized tuning curve of a neuron with a preferred phase near maximal retraction.

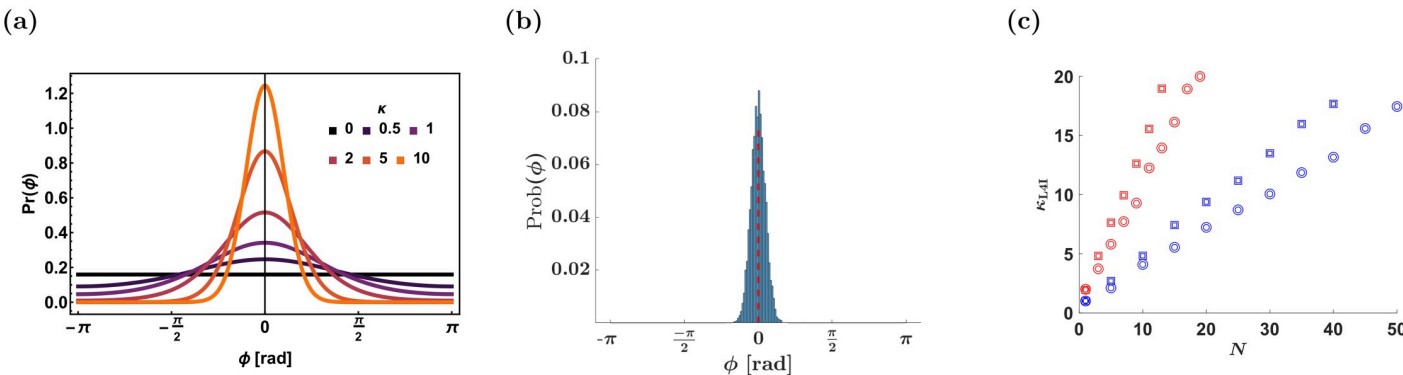

**Fig 2. Distribution of preferred phases.** (a) The von-Mises distribution, Eq (1), with $\psi = 0$ is shown for different values of $\kappa$ as depicted by color. (b) The distribution of preferred phases in the uniform pooling model. The distribution was estimated from 10000 repetitions of drawing $N = 100$ neurons from the upstream population. The preferred phases in the upstream population were drawn in an i.i.d. manner following Eq (1), with $\kappa_{UpStream} = 1$ and $\psi_{UpStream} = 0$. The dashed red line depicts the mean phase. (c) Distribution width in the naive uniform/random pooling model. The width of the distribution of preferred phases, $\kappa_{L4I}$, in the downstream layer (L4I) is shown as a function of the number of pooled VPM neurons, $N$, for the uniform and random pooling in squares and circles, respectively. The width of the distribution, $\kappa_{L4I}$, was estimated from 10000 repetitions of drawing $N$ preferred phases of the upstream population with $\kappa = 1$ (blue) and $\kappa = 2$ (red).

synaptic weights. However, synaptic weights have been reported to be highly volatile [17–20]. If synaptic plasticity is activity independent, then one would expect the synaptic weight profile to become unstructured. In the naive pooling model, which lacks activity-dependent plasticity, we consider two types of unstructured connectivity; namely, uniform pooling and random pooling.

Fig 2b shows the distribution of preferred phases in a naive model that pools the responses of $N = 100$ VPM neurons. The source of variability in the preferred phases of the downstream neuron is the random choice of $N$ VPM neurons, each characterized by a preferred phase that is distributed in an i.i.d. manner following Eq (1). As a result, one would expect that the number, $N$, of pooled thalamic neurons would govern the width of the distribution. Fig 2c shows the expected distribution width, $\kappa_{L4I}$, as a function of $N$ for uniform (squares) and random (circles) pooling. As can be seen from the figure, even a random pooling of $N = 10$ results in a distribution of preferred phases that is considerably narrower than empirically observed. This result is particularly surprising since the number of thalamic neurons synapsing onto a single L4 neuron was estimated to be on the order of 100, see e.g. [16, 21]. Moreover, in the naive pooling model, the mean preferred phase of the downstream population is given by the mean phase of the upstream population. The delay in the response of downstream neurons to their input should also be added. It is estimated to be around 1-10ms; hence, for whisking at 10Hz translates into $\psi_{VPM} - \psi_{L4I} = 0.06\text{-}0.6$ [rad]. This result also runs counter empirical findings reporting $\psi_{VPM} - \psi_{L4I} \approx 2.2$ [rad]. Thus, the synaptic weight profile in a model that lacks activity-dependent plasticity is expected to converge to a random pooling (due to constant synaptic remodelling), which is inconsistent with the observed distribution of preferred phases.

Recently, the effects of rhythmic activity on the spike timing dependent plasticity (STDP) dynamics of feed-forward synaptic connections have been examined [22, 23]. It was shown that in this case the synaptic weights remain dynamic. As a result, the phase of the downstream neuron is arbitrary and drifts in time; thus, effectively, inducing a distribution of preferred phases in the downstream population. However, if the phases of the downstream population are arbitrary and drift in time, how can information about the whisking phase be transmitted?

Here we investigated the hypothesis that the distribution of phases in the downstream layer is governed by the interplay of the distribution in the upstream layer and the STDP rule. The

remainder of this article is organized as follows. First, we define the network architecture and the STDP learning rule. We then derive a mean-field approximation for the STDP dynamics in the limit of a slow learning rate for a threshold-linear Poisson downstream neuron model. Next, we show that STDP dynamics can generate non-trivial distributions of preferred phases in the downstream population and analyze how the parameters characterizing the STDP govern this distribution. Finally, we summarize the results, discuss the limitations of this study and suggest essential empirical predictions deriving from our theory.

## Results

### The upstream thalamic population

We model a system of $N$ thalamic excitatory whisking neurons, synapsing in a feed-forward manner onto a single inhibitory L4 barrel cortical neuron. Unless stated otherwise, following [16], in our numerical simulations we used $N = 150$. The spiking activity of the thalamic neurons is modelled by independent inhomogeneous Poisson processes with an instantaneous firing rate that follows the whisking cycle:

$$\langle \rho_k(t) \rangle = D(1 + \gamma \cos[vt - \phi_k]), \tag{2}$$

where $\rho_k(t) = \sum_i \delta(t - t_{k,i})$, $k \in \{1, \ldots N\}$, is the spike train of the $k$th thalamic neuron, with $\{t_{k,i}\}_{i=1}^{\infty}$ denoting its spike times. The parameter $D$ is the mean firing rate during whisking (averaged over one cycle), $\gamma$ is the modulation depth, $v$ is the angular frequency of the whisking, and $\phi_k$ is the preferred phase of firing of the $k$th thalamic neuron. We further assume that the preferred phases in the thalamic population, $\{\phi_k\}_{k=1}^{N}$, are distributed i.i.d. according to the von-Mises distribution, Eq (1).

### The downstream layer 4 inhibitory neuron model

To facilitate the analysis, we model the response of the downstream layer 4 inhibitory (L4I) neuron to its thalamic inputs by the linear Poisson model, which has been used frequently in the past [22, 24–31]. Given the thalamic responses, the firing of the L4I neuron follows inhomogeneous Poisson process statistics with an instantaneous firing rate

$$r_{\text{L4I}}(t) = \frac{1}{N} \sum_{k=1}^{N} w_k \rho_k(t - d), \tag{3}$$

where $d > 0$ represents the characteristic delay, and $w_k$ is the synaptic weight of the $k$th thalamic neuron.

Due to the rhythmic nature of the thalamic inputs, Eq (2), and the linearity of the L4I neuron, Eq (3), the L4I neuron will exhibit rhythmic activity:

$$\langle r_{\text{L4I}}(t) \rangle = D_{\text{L4I}}(1 + \gamma_{\text{L4I}} \cos[vt - \psi_{\text{L4I}}]), \tag{4}$$

with a mean, $D_{\text{L4I}}$, a modulation depth, $\gamma_{\text{L4I}}$, and a preferred phase $\psi_{\text{L4I}}$, that depend on global order parameters that characterize the thalamocortical synaptic weights population. For large $N$ these order parameters are given by:

$$\bar{w}(t) = \int_0^{2\pi} \Pr(\phi) w(\phi, t) d\phi \tag{5}$$

and

$$\tilde{w}(t)e^{i\psi} = \int_0^{2\pi} \Pr(\phi)w(\phi,t)e^{i\phi}d\phi. \tag{6}$$

where $\bar{w}$ is the mean synaptic weight and $\tilde{w}e^{i\psi}$ is its first Fourier component. The phase $\psi$ is determined by the condition that $\tilde{w}$ is real and non-negative. Consequently, the L4I neurons in our model respond to whisking with a mean $D_{\mathrm{L4I}} = D\bar{w}$, a modulation depth of $\gamma_{\mathrm{L4I}} = \gamma\tilde{w}/\bar{w}$, and a preferred phase $\psi_{\mathrm{L4I}} = \psi + \nu d$.

## The STDP rule

We model the modification of the synaptic weight, $\Delta w$, following either a pre- or post-synaptic spike as the sum of two processes: potentiation (+) and depression (-) [31–33], as

$$\Delta w = \lambda[f_+(w)K_+(\Delta t) - f_-(w)K_-(\Delta t)]. \tag{7}$$

The parameter $\lambda$ denotes the learning rate. We further assume a separation of variables and write each term (potentiation and depression) as the product of the function of the synaptic weight, $f_\pm(w)$, and the temporal kernel of the STDP rule, $K_\pm(\Delta t)$, where $\Delta t = t_{\mathrm{post}} - t_{\mathrm{pre}}$ is the time difference between pre- and post-synaptic spike times. Following Gütig et al. [33] the weight dependence functions, $f_\pm(w)$, were chosen to be:

$$f_+(w) \quad = (1 - w)^\mu \tag{8a}$$

$$f_-(w) \quad = \alpha w^\mu, \tag{8b}$$

where $\alpha > 0$ is the relative strength of depression and $\mu \in [0, 1]$ controls the non-linearity of the learning rule.

The temporal kernels of the STDP rule are normalized: i.e., $\int K_\pm(\Delta t)d\Delta t = 1$. Here, for simplicity, we assume that all pairs of pre and post spike times contribute additively to the learning process via Eq (7).

Empirical studies portray a wide variety of temporal kernels [34–42]. Specifically, in our work, we used two families of STDP rules: 1. A temporally asymmetric kernel [35, 37–39]; 2. A temporally symmetric kernel [38, 40–42]. Both of these rules have been observed in the the barrel system of mice, at least for some developmental period [43–45]. For the temporally asymmetric kernel we used the exponential model,

$$K_\pm(\Delta t) = \frac{e^{\mp\Delta t/\tau_\pm}}{\tau_\pm}\Theta(\pm\Delta t), \tag{9}$$

where $\Theta(x)$ is the Heaviside function, and $\tau_\pm$ are the characteristic timescales of the potentiation (+) or depression (−). We take $\tau_- > \tau_+$ as typically reported.

For the temporally symmetric learning rule we used a difference of Gaussians model,

$$K_\pm(\Delta t) = \frac{1}{\tau_\pm\sqrt{2\pi}}e^{-\frac{1}{2}\left(\frac{\Delta t}{\tau_\pm}\right)^2}, \tag{10}$$

where $\tau_\pm$ are the temporal widths. In this case, the order of firing is not important, only the absolute time difference.

## STDP dynamics in the limit of slow learning

In the limit of a slow learning rate, $\lambda \to 0$, we obtain deterministic dynamics for the mean synaptic weights (see [32] for a detailed derivation)

$$\frac{\dot{w}_j(t)}{\lambda} = I_j^+(t) - I_j^-(t) \tag{11}$$

with

$$I_j^\pm(t) = f_\pm(w_j(t)) \int_{-\infty}^{\infty} \Gamma_{j,\text{post}}(\Delta) K_\pm(\Delta) d\Delta, \tag{12}$$

where $\Gamma_{j,\text{L4I}}(\Delta)$ is the cross-correlation function between the $j$th thalamic pre-synaptic neuron and the L4I post-synaptic neuron; see detailed derivation in Temporal correlations.

## STDP dynamics of thalamocortical connectivity

We simulated the STDP dynamics of 150 upstream thalamic neurons synapsing onto a single L4I neuron in the barrel cortex, see Details of the numerical simulations & statistical analysis.

Fig 3a shows the temporal evolution of the synaptic weights (color-coded by their preferred phases). As can be seen from the figure, the synaptic weights do not relax to a fixed point; instead there is a continuous remodelling of the entire synaptic population. Examining the order parameters, Fig 3b and 3c, reveals that the STDP dynamics converge to a limit cycle.

The continuous remodelling of the synaptic weights causes the preferred phase of the downstream neuron, $\kappa_{\text{L4I}}$ (see Eq (4)) to drift in time, Fig 3b. As can be seen from the figure,

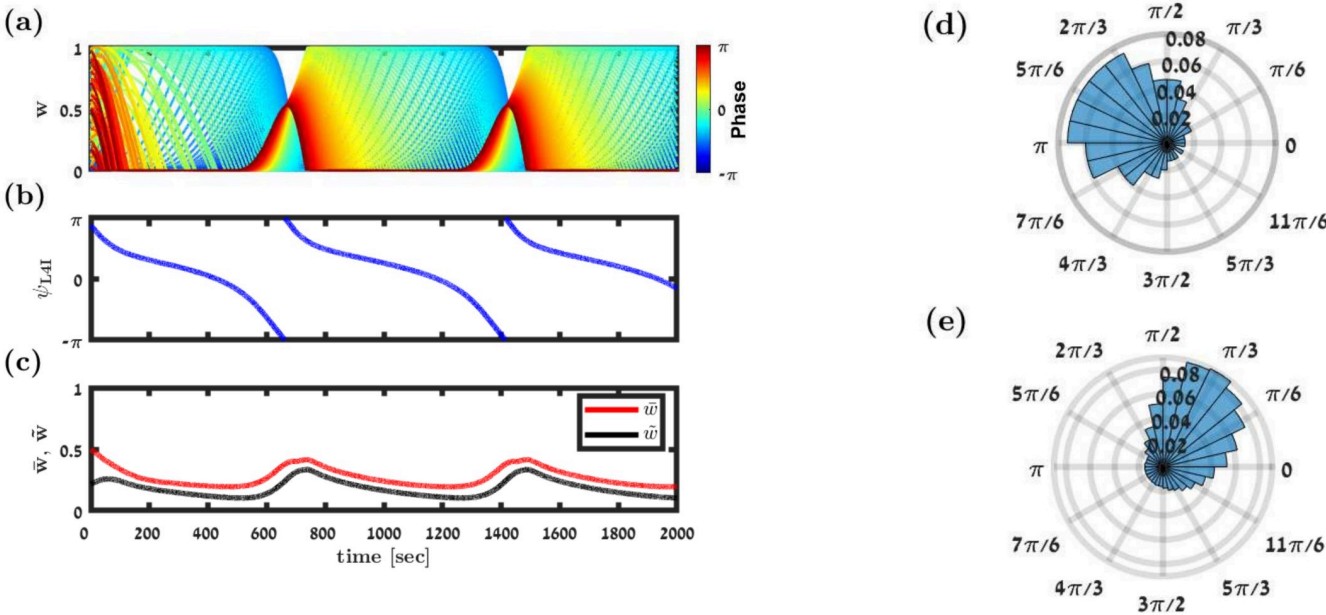

**Fig 3. Simulation of the STDP dynamics.** (a) Synaptic weight dynamics. Each trace depicts the time evolution of a single synaptic weight, differentiated by color according to its preferred phase, see legend. (b) & (c) Dynamics of the order parameters. The preferred phase of the downstream neuron, $\kappa_{\text{L4I}}$, (in (b)), and the mean, $\bar{w}$, and the magnitude of the first Fourier component, $\tilde{w}$, (in red and black, respectively, in (c)) are shown as a function of time. (d) The distribution of preferred phases in the thalamic population that served as input to the downstream L4I neuron is presented as a polar histogram. The distribution followed Eq (1) with $\kappa_{\text{VPM}} = 1$ and $\psi_{\text{VPM}} = 5\pi/6$ rad. (e) The temporal distribution of preferred phases of the downstream L4I neuron is shown in a polar plot. Fitting the von-Mises distribution yielded $\kappa_{\text{L4I}} = 1.1$ and $\kappa_{\text{L4I}} = 0.8$ rad. In this simulation we used the following parameters: $N = 150$, $\bar{v} = v/(2\pi) = 7$hz, $\gamma_{\text{VPM}} = 1$, $D_{\text{VPM}} = 10$hz, and $d = 3$ms. The temporally asymmetric STDP rule, Eq (9), was used with $\tau_- = 50$ms, $\tau_+ = 22$ms, $\mu = 0.01$, $\alpha = 1.1$, and $\lambda = 0.01$.

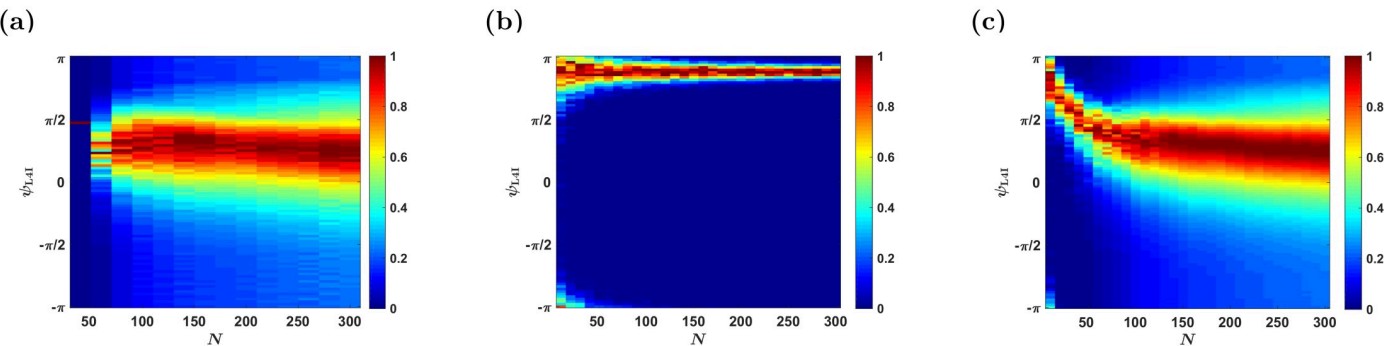

**Fig 4. The effects of population size.** The distribution of the preferred phases of L4I neurons, $\kappa_{L4I}$, is shown as a function of $N$ for different sources of variability. In each column (value of $N$), the non-normalized distribution: $\Pr(\psi)/\max\{\Pr(\psi)\}$, is presented by color. (a) The temporal distribution of the preferred phase of a single L4I neuron as a result of STDP dynamics without quenched disorder or averaging (see Modeling pre-synaptic phase distributions) is presented. (b) Distribution due to quenched disorder without STDP. The distribution of L4I neurons in the randomly pooling model was estimated from 1000 trials of drawing the preferred phases of $N$ VPM neurons in an *i.i.d.* manner from Eq (1). (c) The distribution due to both quenched disorder and STDP dynamics is shown. Unless stated otherwise, the parameters used in these graphs are as follows: $\bar{v} = v/(2\pi) = 7$Hz, $\kappa_{VPM} = 1$, $\gamma = 0.9$, $D = 10$hz, $d = 3$ms, $\phi_0 = 5\pi/6$. For the STDP we used the temporally exponential asymmetric kernel, Eq (9), with $\tau_- = 50$ms, $\tau_+ = 22$ms, $\mu = 0.01$, $\alpha = 1.1$, and $\lambda = 0.01$.

the drift velocity is not constant. Consequently, the downstream (L4I) neuron 'spends' more time in certain phases than others. Thus, the STDP dynamics induce a distribution over time for the preferred phases of the downstream neuron. One can estimate the distribution of the preferred phases of L4I neurons by tracking the phase of a single neuron over time. Alternatively, since our model is purely feed-forward, the preferred phases of different L4I neurons are independent; hence, this distribution can also be estimated by sampling the preferred phases of different L4I neurons at the same time. Fig 3d and 3e show the distribution of preferred phases of thalamic and L4I whisking neurons, respectively. Thus, STDP induces a distribution of preferred phases in the L4I population, which is linked to the temporal distribution of single L4I neurons.

Fig 4a shows the (non-normalized) distribution of preferred phases induced by STDP as a function of the number of pooled thalamic neurons, $N$. For large $N$, the STDP dynamics converge to the continuum limit of Eq (11), and the distribution converges to a limit that is independent of $N$. This contrasts with the naive pooling model lacking plasticity (cf. Fig 4b). In that model, the distribution of preferred phases in the cortical population results from a random process of pooling phases from the thalamic population. We shall refer to this type of variability as quenched disorder, since this randomness is frozen and does not fluctuate over time. This process is characterized by a distribution of preferred phases with vanishing width, $\kappa_{L4I} \to \infty$, in the large $N$ limit, Fig 4b. In addition to the different widths of the distribution, the mean of the preferred phases also differs greatly. Whereas in Fig 4a the mean preferred phase is determined by the STDP rule, without STDP, Fig 4b, the mean phases of L4I neurons is given by the mean preferred phase of the VPM population shifted by the delay, $\kappa_{L4I} = \psi_{VPM} + dv$.

For small $N$, $N \lesssim 70$, STDP induces a point measure distribution over time, Fig 4a. This is due to pinning in the noiseless STDP dynamics in the mean field limit, $\lambda \to 0$. Stochastic dynamics, due to noisy neuronal responses, can overcome pinning. Fig 4c depicts the distribution of the preferred phases for different values of $N$, which results from both STDP dynamics and quenched averaging over different realizations of the preferred phases of the thalamic neurons. For small values of $N$, the distribution is dominated by the quenched statistics, in terms of its narrow width and mean that is dominated by the mean phase of the thalamic population.

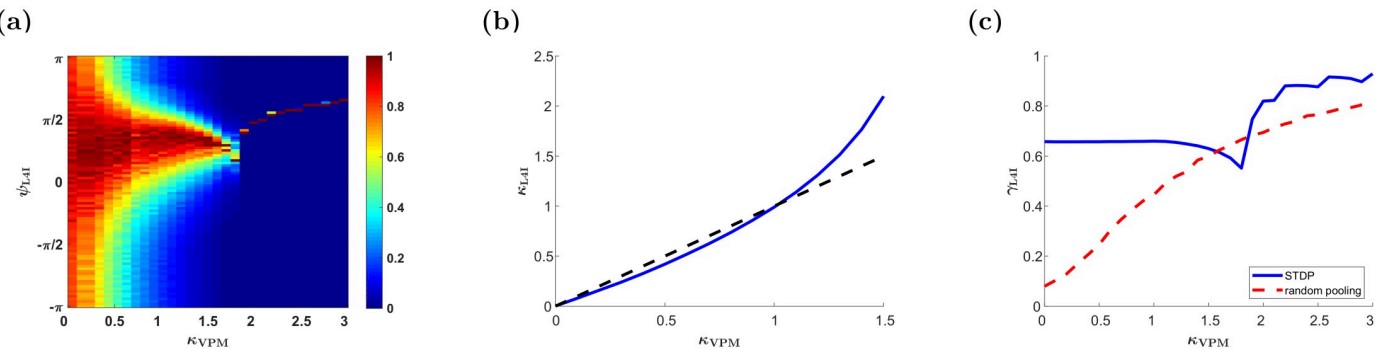

**Fig 5. The effect of the distribution width of the upstream population, $\kappa_{\mathrm{VPM}}$.** (a) The (non-normalized) distribution of the preferred phases of L4I neurons, $\kappa_{\mathrm{L4I}}$, is shown by color as a function of the width of the distribution of preferred phases in VPM, $\kappa_{\mathrm{VPM}}$. (b) The distribution width in layer 4, $\kappa_{\mathrm{L4I}}$, is shown as a function of $\kappa_{\mathrm{VPM}}$ (blue). The identity line is shown (dashed black) for comparison. (c) The modulation depth in the downstream population, $\gamma_{\mathrm{L4I}}$, is shown as a function of $\kappa_{\mathrm{VPM}}$ (blue). For comparison, the modulation depth of the uniform pooling model is also presented (dashed red). The parameters used here are: $\bar{v} = v/(2\pi) = 7$Hz, $\kappa_{\mathrm{VPM}} = 1$, $\gamma = 1$, $D = 10$hz, $d = 3$ms, $\phi_0 = 5\pi/6$. For the STDP we used the temporally exponential asymmetric kernel, Eq (9), with $\tau_- = 50$ms, $\tau_+ = 22$ms, $\mu = 0.01$, $\alpha = 1.1$, and $\lambda = 0.01$.

As $N$ increases, the distribution widens and is centered around a preferred phase that is determined by the STDP. Thus, activity dependent plasticity helps to shape the distribution of preferred phases in the downstream population. Below, we examine how these different parameters affect the ways in which STDP shapes the distribution; hence, we will not average over the quenched disorder.

**Parameters characterizing the upstream input.** Fig 5a depicts the distribution of preferred phases as a function of the distribution width of their thalamic input, $\kappa_{\mathrm{VPM}}$. For a uniform input distribution, $\kappa_{\mathrm{VPM}} = 0$, the downstream distribution is also uniform, $\kappa_{\mathrm{L4I}} = 0$, see [22]. As the distribution in the VPM becomes narrower, so does the distribution in the L4I population. If the distribution of the thalamic population is narrower than a certain critical value, STDP will converge to a fixed point and $\kappa_{\mathrm{L4I}}$ will diverge. Typically, we find that the width of the cortical distribution, $\kappa_{\mathrm{L4I}}$, is similar to or larger than that of the upstream distribution, $\kappa_{\mathrm{VPM}}$, Fig 5b. This sharpening is obtained via STDP by selectively amplifying certain phases while attenuating others. Consequently the rhythmic component, in terms of the modulation depth, $\gamma_{\mathrm{L4I}}$, is also typically amplified relative to the uniform pooling model, Fig 5c.

The effect of the whisking frequency is shown in Fig 6a. For moderate rhythms that are on a similar timescale to that of the STDP rule, STDP dynamics can generate a wide distribution of preferred phases. However, in the high frequency limit, $v \to \infty$, the synaptic weights converge to a uniform solution with $w(\phi) = (1 + \alpha^{1/\mu})^{-1}, \forall \phi$ (see [22]). In this limit, due to the uniform pooling, there is no selective amplification of phases and the whisking signal is transmitted downstream due to the selectivity of the thalamic population, $\kappa_{\mathrm{VPM}} > 0$. Consequently, at high frequencies, the distribution of preferred phases will be extremely narrow, $\mathcal{O}(1/\sqrt{N})$ because of the quenched disorder, and the rhythmic signal will be attenuated, $\gamma_{\mathrm{L4I}} = \gamma_{\mathrm{VPM}} \times I_1(\kappa)/I_0(\kappa)$ (where $I_j(\kappa)$ is the modified Bessel function of order $j$), Fig 6b. The rate of convergence to the high frequency limit is governed by the smoothness of the STDP rule. Discontinuity in the STPD rule, such as in our choice of a temporally asymmetric rule, will induce algebraic convergence in $v$ to the high frequency limit, whereas a smooth rule, such as our choice of a temporally symmetric rule, will manifest in exponential convergence; compare Fig 6 with Fig 7, see also [22]. In our choice of parameters, $\tau_+ \approx 20$ms and $\tau_- \approx 50$ms, STDP dynamics induced a

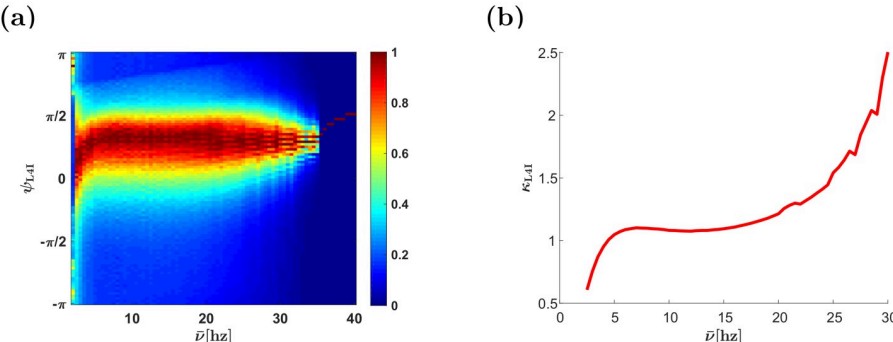

**Fig 6. Effects of whisking frequency.** (a) The (non-normalized) distribution of L4I neuron phases, $\kappa_{L4I}$, is depicted by color as a function, $\bar{v} = v/(2\pi)$. (b) The width of the distribution, $\kappa_{L4I}$, of L4I neurons is shown as a function of $\bar{v}$. The parameters used in these graphs were: $\kappa_{VPM} = 1$, $\psi_{VPM} = 5\pi/6$ $\gamma = 0.9$, $D = 10$hz, and $d = 3$ms. We used the temporally asymmetric exponential learning rule, Eq (9), with $\tau_- = 50$ms, $\tau_+ = 22$ms, $\mu = 0.01$, $\alpha = 1.1$, and $\lambda = 0.01$.

wide distribution for frequencies in the $\alpha$, $\beta$ and low $\gamma$ bands in the case of the asymmetric learning rule Fig 6, and around 5–15hz in the case of the symmetric learning rule Fig 7.

**The effects of synaptic weight dependence, $\mu$.** Previous studies have shown that increasing $\mu$ weakens the positive feedback of the STDP dynamics, which generates multi-stability, and stabilizes more homogeneous solutions [22, 23, 33, 46]. This transition is illustrated in Fig 8: at low values of $\mu$, the STDP dynamics converge to a limit cycle in which both the synaptic weights and the phase of the the L4I neuron cover their entire dynamic range, Fig 8a and 8b. As $\mu$ is increased, the synaptic weights become restricted in the limit cycle and no longer span their entire dynamic range, Fig 8c and 8d. A further increase of $\mu$ also restricts the phase of the L4I neuron along the limit cycle, Fig 8e and 8f. Finally, when $\mu$ is sufficiently large, STDP dynamics converge to a fixed point, Fig 8g and 8h. This fixed point selectively amplifies certain phases, yielding a higher value of $\gamma$ than in the uniform solution. These results are summarized in Fig 9 that shows the (non-normalized) distribution of preferred phases and the relative strength of the rhythmic component, $\gamma_{L4I}$, as a function of $\mu$. Note that except for a small value

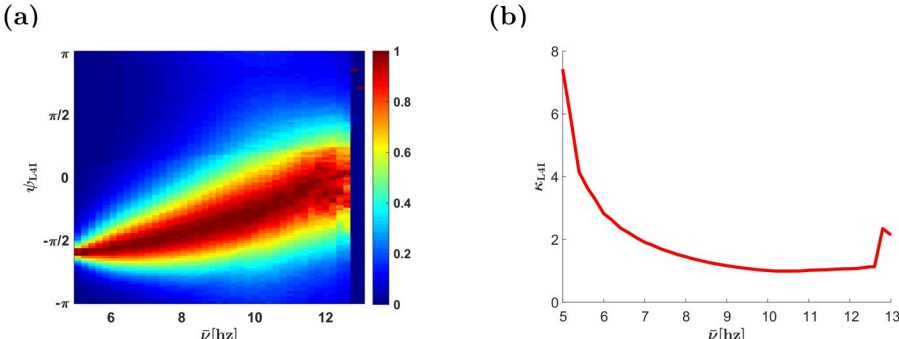

**Fig 7. Effects of whisking frequency—The temporally symmetric STDP rule.** (a) The (non-normalized) distribution of L4I neuron phases, $\kappa_{L4I}$, is depicted by color as a function, $\bar{v} = v/(2\pi)$. (b) The width of the distribution, $\kappa_{L4I}$, of L4I neurons, is shown as a function of $\bar{v}$. Unless stated otherwise, the parameters used here were: $\kappa_{VPM} = 1$, $\psi_{VPM} = 5\pi/6$, $\gamma = 0.9$, $D = 10$hz, and $d = 10$ms. We used the temporally asymmetric exponential learning rule, Eq (10), with: $\tau_- = 50$ms, $\tau_+ = 22$ms, $\mu = 0.01$, $\alpha = 1.1$, and $\lambda = 0.01$.

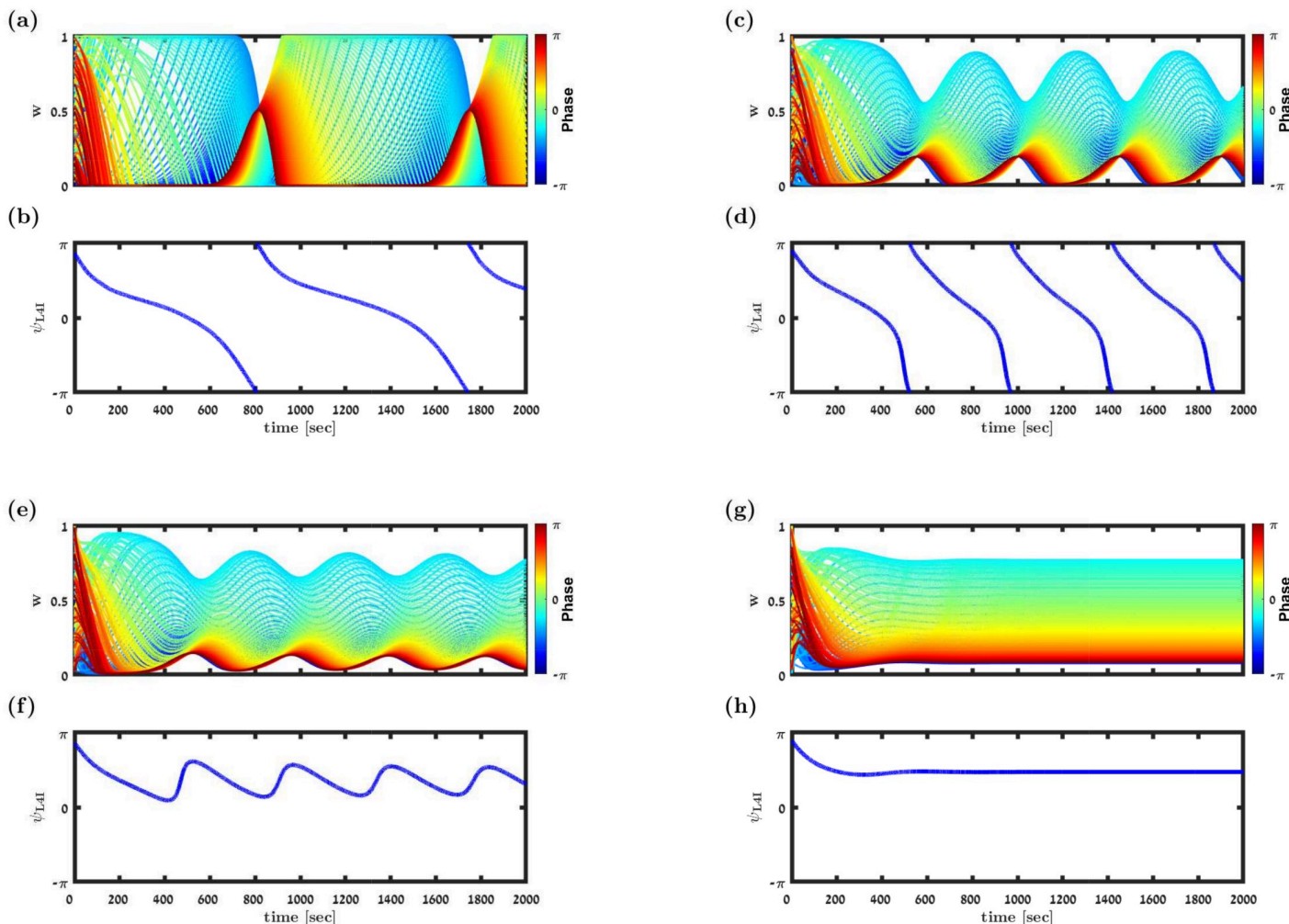

**Fig 8. Transition to a fixed point solution.** (a), (c), (e) & (f) show the synaptic weight dynamics for different values of $\mu$. The synaptic weights are depicted by different traces, colored according to the preferred phase of the pre-synaptic neuron, see legend. (b), (d), (f) & (h) show the preferred phase of the downstream L4I neuron, $\kappa_{L4I}$, as a function of time. The parameters used in these simulation were: $N = 150$, $\kappa = 1$, $\bar{v} = v/(2\pi) = 7$Hz, $\gamma = 0.9$, $D = 10$Hz, and $d = 3$ms. We used here the temporally asymmetric learning rule, Eq (9), with: $\tau_- = 50$ms, $\tau_+ = 22$ms, and $\alpha = 1.1$. For the non-linearity parameter we used: $\mu = 0.01$ in (a) & (b), $\mu = 0.06$. in (c) & (d), $\mu = 0.07$ in (e) & (f). and $\mu = 0.1$ in (g) & (h).

range of $\mu$, around which the distribution of preferred phases is bi-modal (see, e.g. Fig 8e and 8f and $\mu \approx 0.07$ in Fig 9), STDP yields higher $\gamma_{L4I}$ values than in the uniform pooling model.

**The relative strength of depression, $\alpha$.** As one might expect, decreasing $\alpha$ beyond a certain value will result in a potentiation dominated STDP dynamics, saturating all synapses close to their maximal value, thus, approaching the uniform solution, which is characterized by a narrow preferred phase distribution centered around the mean preferred phase of the VPM neurons shifted by the delay, and low values of $\gamma_{L4I}$, Fig 10. Increasing $\alpha$ strengthens the competitive winner-take-all like nature of the STDP dynamics. Initially, this competition will generate a fixed point with non-uniform synaptic weights, thus increasing $\gamma_{L4I}$. A further increase of $\alpha$ results in a limit cycle solution to the STDP dynamics that widens the distribution of the preferred phases. Increasing $\alpha$ beyond a certain critical value will result in depression dominated STDP dynamics, thus driving the synaptic weights to zero, Fig 10.

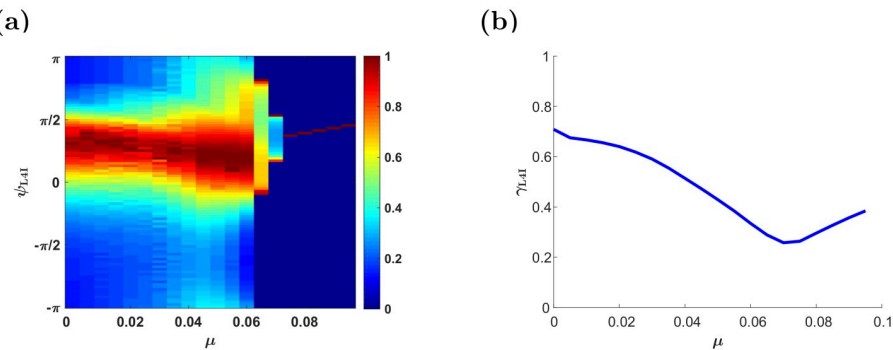

**Fig 9. The effect of the non-linearity parameter, $\mu$.** (a) The distribution of preferred phases of L4I neurons, $\kappa_{L4I}$, is shown by color as a function of $\mu$. (b) The modulation depth, $\gamma_{L4I}$, is depicted as a function of $\mu$. The parameters used here were: $\kappa_{VPM} = 1$, $\psi_{VPM} = 5\pi/6$, $\gamma = 0.9$, $D = 10hz$, and $d = 3ms$. The temporally asymmetric STDP rule, Eq (9), was used with: $\tau_- = 50ms$, $\tau_+ = 22ms$, $\alpha = 1.1$ and $\lambda = 0.01$.

**Parameters characterizing the temporal structure of STDP.** Fig 11 shows the effect of the temporal structure of the STDP on the distribution of preferred phases in the downstream population. As can be seen from the figure, varying the characteristic timescales of potentiation and depression, $\tau_+$ and $\tau_-$, induces quantitative effects on the distribution of the preferred phases. However, qualitative effects result from changing the nature of the STDP rule. Comparing the temporally asymmetric rule, Fig 11a and 11b, to the temporally symmetric rule, Fig 11c and 11d, reveals a dramatic difference in the mean preferred phase of L4I neurons, dashed black lines.

## Discussion

We examined the possible contribution of STDP to the diversity of preferred phases of whisking neurons. Whisking neurons can be found along different stations throughout the somatosensory information processing pathway [1, 4–6, 9, 10, 13, 16, 47]. Here we focused on L4I whisking neurons that receive their whisking input mainly via excitatory inputs from the VPM, and suggested that the non-trivial distribution of preferred phases of L4I neurons results from a continuous process of STDP.

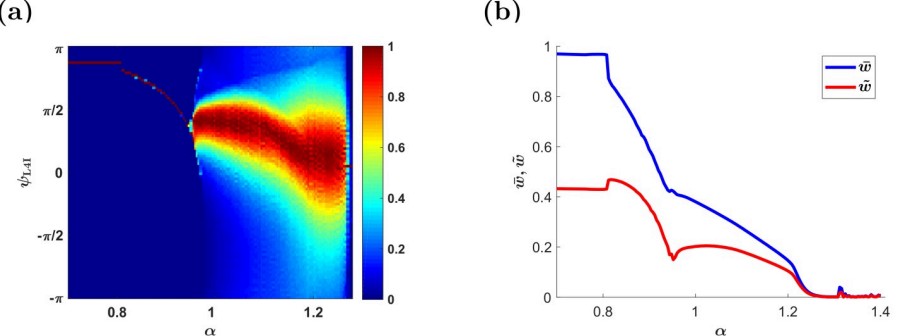

**Fig 10. The effect of the relative strength of depression, $\alpha$.** (a) The mean input phase $\kappa_{L4I}$ as a function of $\alpha$. Probability of occurrences is depicted by color as shown in the right color bar. (b) The order parameters $\bar{w}$ and $\tilde{w}$ are shown as a function of $\alpha$. Here, we used: $\kappa_{VPM} = 1$, $\psi_0 = 5\pi/6$, $\gamma = 1$, $D = 10hz$, and $d = 3ms$. The temporally asymmetric STDP rule, Eq (9), was used with: $\tau_- = 50ms$, $\tau_+ = 22ms$, $\mu = 0.01$ and $\lambda = 0.01$.

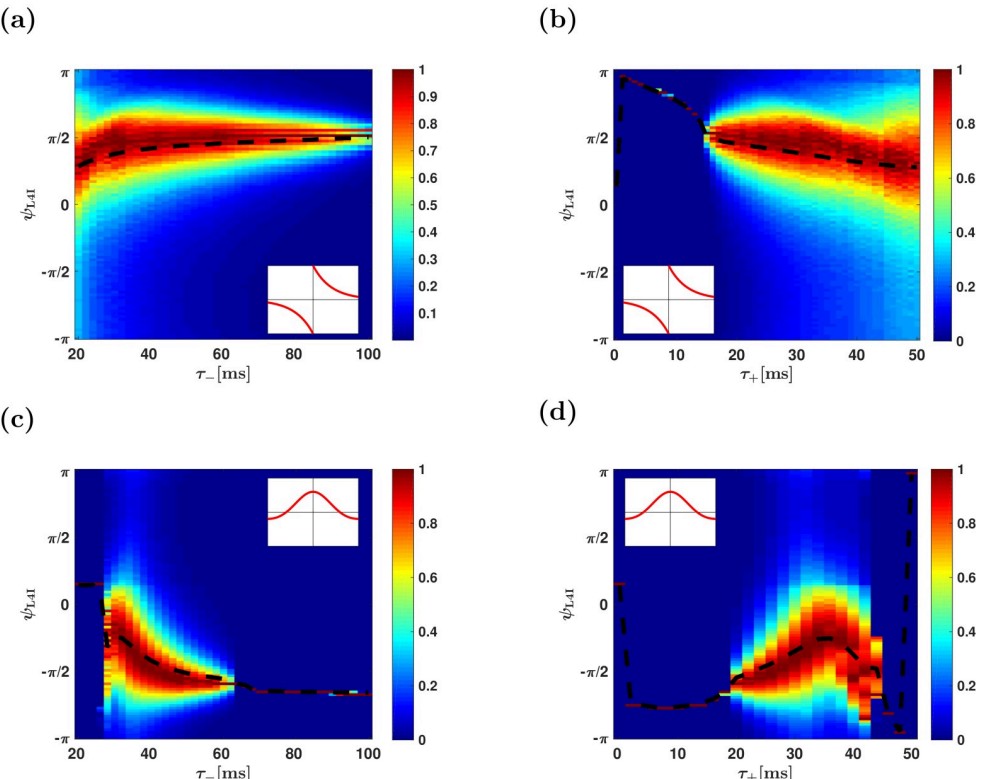

**Fig 11. The effect of the temporal structure of the STDP rule.** The distribution of preferred phases of L4I neurons, $\kappa_{\text{L4I}}$, is shown as a function of the characteristic timescales of the STDP: $\tau_-$ in (a) & (c) and $\tau_-$ + in (b) & (d), for the temporally asymmetric rule in (a) & (b) and the symmetric rule in (c) & (d). The dashed black lines depict the mean phase. Unless stated otherwise, the parameters used here were: $\kappa_{\text{VPM}} = 1$, $\gamma = 1$, $D = 10$hz, $\tau_- = 50$ms, $\tau_+ = 22$ms, $d = 10$ms, $\psi_{\text{VPM}} = 5\pi/6$, $\mu = 0.01$, $\alpha = 1.1$ and $\lambda = 0.01$.

STDP has been reported in thalamocortical connections in the barrel system. However, STDP has only been observed during the early stages of development [43–45]. Is it possible that the contribution of STDP to shaping thalamocortical connectivity is restricted to the early developmental stages? This is an empirical question that can only be answered experimentally. Nevertheless, several comments can be made.

First, Inglebert and colleagues recently showed that pairing single pre- and post-synaptic spikes, under normal physiological conditions, does not induce plastic changes [48]. On the other hand, activity dependent plasticity was observed when stronger activity was induced. Thus, whisking activity, which is a strong collective and synchronized activity, may induce STDP of thalamocortical connectivity.

Second, in light of the considerable volatility of synaptic connections observed in the central nervous system [17–20], it is hard to imagine that thalamocortical connectivity remains unchanged throughout the lifetime of an animal. Third, if activity independent plasticity alone underlies synaptic volatility, thalamocortical synaptic weights would be expected to be random. In this case, thalamic whisking input to layer 4 should be characterized by an extremely narrow distribution centered around the delayed mean thalamic preferred phase, cf. Fig 4b. Consequently, neglecting the possible contributions due to recurrent interactions with layer 4, a non-trivial distribution of L4I phases, with $\kappa_{\text{L4I}} \sim 1$, can be obtained either via STDP or by pooling the whisking signal from an extremely small VPM population of $N < 10$ neurons.

However, in the latter scenario (without STDP) the mean preferred phase of L4I neurons is expected to be determined by the (delayed) mean phase of VPM neurons (see e.g. Fig 4b), thus raising serious doubts as to the viability of the latter solution.

An alternative hypothesis to our proposed mechanism is that the recurrent dynamics within layer 4 generate the empirically observed wide distribution of preferred phases centered on $\kappa_{L4I} \approx \psi_{VPM} - 2.2$ [rad]. As layer 4 excitatory neurons have been reported to exhibit very weak rhythmic activity [13, 16], such a mechanism would mainly be based on recurrent interaction within the L4I population. The putative ways in which inhibitory interactions widen the distribution remain unclear. Nevertheless, this hypothesis should be explored further.

Our hypothesis views STDP as a continuous process. As a result of STDP, the preferred phase of a L4I neuron, in our model, is a stochastic process with a non-uniform drift, Fig 3b, which, in turn, induces a distribution over time for the phase, Fig 3e. As our model lacks recurrent interactions, phases of different L4I neurons will be i.i.d. random processes, albeit with different random initial conditions. Consequently, simultaneous sampling of the distribution of preferred phases of a population of L4I neurons will yield the same distribution as obtained by sampling the preferred phase of a single neuron over time. Thus, functionality, in terms of transmission of the whisking signal and retaining the stationary distribution of the preferred phases in the downstream population is obtained as a result of continuous remodelling of the entire population of synaptic weights.

Several key features of our hypothesis provide clear empirical predictions. Our theory links two empirical observations. One is the distribution of preferred phases in up- and downstream populations. The other is the STDP rule. The literature can provide a good assessment of the distribution of preferred phases of thalamic and of cortical whisking neurons. Further effort is required to estimate the STDP rule of the thalamocortical connections to L4I neurons. Thus, we can reject our hypothesis if the predicted distribution, based on our theory and the estimated learning rule, is not consistent with the empirically observed distribution.

Our theory also posits that the distribution of preferred phases results from their stochastic dynamics. Our theory also incorporates a direct relationship between the drift velocity and the distribution of preferred phases. Essentially, the system will spend more time around phases characterized by slower velocity; hence, a higher probability of preferred phases should correlate with slower drift velocity. To the best of our knowledge, the dynamics of the preferred phases of whisking neurons has not been studied. Monitoring the preferred phases of single L4I neurons over long periods of time (days and weeks) could serve as additional tests of our theory. Specifically, it would address the question of whether the preferred phases drift in time, and if so, whether the drift velocity and the distribution of preferred phases are consistent with our theory.

Further empirical exploration can test the degree in which our approximation of a purely feed-forward architecture (i.e., neglecting recurrent interactions) holds. Simultaneous recordings of a large population of fast spiking layer 4 whisking neurons can be harnessed to compare the distribution of preferred phases over space and time, which are expected to be identical in the absence of recurrent interactions. In addition, in the visual system, Lien and Scanziani combined intracellular recordings with optogenetic methods to assess the contribution of thalamic inputs to the selectivity of cortical neurons [49]. In the context of the current work, we explored the thalamic inputs to whisking neurons in the barrel cortex. Thus, applying ideas and approaches from the work of Lien and Scanziani could (*i*) provide a measurement of the thalamic input, and (*ii*) determine the relative contribution of feed-forward and recurrent connections to the selectivity to the phase of whisking in L4I neurons.

More broadly, our findings raise the question of whether the constant remodelling of synaptic efficacies is an artefact of the 'biological hardware' the brain must overcome, or whether it reflects an underlying principle of the central nervous system. Though we cannot provide an

answer to this question at this point in time, we may speculate on the possible advantages of a dynamic solution. One interesting hypothesis to consider is that the constant remodelling makes the neuronal network more flexible and consequently enables a faster response to a changing environment.

In the current study, we made several simplifying assumptions. The spiking activity of the thalamic population was modeled as a rhythmic activity with a well-defined frequency. However, whisking activity spans a frequency band of several Hertz. Moreover, the thalamic input relays additional signals, such as touch and texture. These signals modify the cross-correlation structure and add 'noise' to the dynamics of the preferred phase of the downstream neuron. As a result, the distribution of preferred phases in the downstream population is likely to be wider. In addition, our analysis used a purely feed-forward architecture ignoring recurrent connections in layer 4, which may also affect the preferred phases in layer 4. A quantitative application of our theory to the whisker system should consider all these effects. Nevertheless, the theory presented provides the basic foundation to address these effects.

## Methods

### Temporal correlations

The cross-correlation between pre-synaptic neurons $j$ and $k$ at time difference $\Delta t$ is given by:

$$\Gamma_{(j,k)}(\Delta t) = \langle \rho_j(t) \rho_k(t+\Delta t) \rangle = D^2\left(1 + \frac{\gamma^2}{2}\cos[\nu \Delta t + \phi_j - \phi_k]\right) + \delta_{jk}D\delta(\Delta t).$$ 

(13)

In the linear Poisson model, Eq (3), the cross-correlation between a pre-synaptic neuron and the post-synaptic neuron can be written as a linear combination of the cross-correlations in the upstream population; hence, the cross-correlation between the $j$th VPM neuron and the post-synaptic neuron is

$$\Gamma_{j,\,\text{post}}(\Delta t) = \frac{D}{N}\delta(\Delta t - d)w_j + D^2\left(\bar{w} + \frac{\gamma^2}{2}\tilde{w}\cos[\nu(\Delta t - d) + \phi_j - \psi]\right).$$ 

(14)

Where $\bar{w}$ and $\tilde{w}e^{i\psi}$ are order parameters characterizing the synaptic weights profile, as defined in Eqs (5) and (6).

### The mean field Fokker-Planck dynamics

For large $N$ we obtain the continuum limit from Eq (11):

$$\frac{\dot{w}(\phi, t)}{\lambda} = F_d(\phi, t) + \bar{w}(t)F_0(\phi, t) + \tilde{w}(t)F_1(\phi, t),$$ 

(15)

where

$$F_d(\phi, t) = w(\phi, t)\frac{D}{N}\left(f_+(w(\phi, t))K_+(d) - f_-(w(\phi, t))K_-(d)\right),$$ 

(16a)

$$F_0(\phi, t) = D^2 \left( \bar{K}_+ f_+(w(\phi, t)) - \bar{K}_- f_-(w(\phi, t)) \right), \tag{16b}$$

$$
\begin{aligned}
F_1(\phi, t) = D^2 \frac{\gamma^2}{2} \Big( &\tilde{K}_+ f_+(w(\phi, t)) \cos[\phi - \Omega_+ - \\
&vd - \psi] - \tilde{K}_- f_-(w(\phi, t)) \cos[\phi - \Omega_- \\
&-vd - \psi] \Big),
\end{aligned}
\tag{16c}
$$

and $\bar{K}_\pm$, $\tilde{K}_\pm e^{i\Omega_\pm^\eta}$ are the Fourier transforms of the STDP kernels

$$\bar{K}_\pm = \int_{-\infty}^{\infty} K_\pm(\Delta) d\Delta, \tag{17}$$

$$\tilde{K}_\pm e^{i\Omega_\pm} = \int_{-\infty}^{\infty} K_\pm(\Delta) e^{-iv\Delta} d\Delta. \tag{18}$$

Note that in our specific choice of kernels, $\bar{K}_\pm = 1$, by construction.

## Fixed points of the mean field dynamics

The fixed point solution of Eq (15) is given by

$$w(\phi)^* = \left( 1 + \alpha^{1/\mu} \left( \frac{1 + X_-}{1 + X_+} \right)^{1/\mu} \right)^{-1}, \tag{19}$$

where

$$X_\pm \equiv \frac{\tilde{w}}{\bar{w}} \frac{\gamma^2}{2} \tilde{K}_\pm \cos(\phi - vd - \Omega_\pm - \psi). \tag{20}$$

Note that, from Eqs (19) and (20), the fixed point solution, $w(\phi)^*$, will depend on $\phi$, for $\kappa_{\mathrm{VPM}} > 0$. As $\mu$ grows to 1 the fixed point solution will become more uniform, see [22].

## Details of the numerical simulations & statistical analysis

Scripts of the numerical simulations were written in Matlab. The numerical results presented in this paper were obtained by solving Eq (15) with the Euler method with a time step $\Delta t = 0.1$ and $\lambda = 0.01$. The cftool with the Nonlinear Least Squares method and a confidence level of 95% for all parameters was used for fitting the data presented in Figs 3, 5b, 6b and 7b.

## Modeling pre-synaptic phase distributions

Unless stated otherwise, STDP dynamics in the mean field limit was simulated without quenched disorder. To this end, the preferred phase, $\phi_k$, of the $k$th neuron in a population of $N$ pre-synaptic VPM neurons was set by the condition $\int_{-\pi}^{\varphi_k} \Pr(\varphi) d\phi = k/N$. In Fig 4b and 4c we used the accept-reject method [50, 51] to sample the phases.

## Author Contributions

**Conceptualization:** Maoz Shamir.

**Investigation:** Nimrod Sherf, Maoz Shamir.

**Software:** Nimrod Sherf.

**Writing – original draft:** Nimrod Sherf, Maoz Shamir.

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
