## [Decision Letter · Decision Letter 0]

12 Jul 2021

Dear Mr. Sherf,

Thank you very much for submitting your manuscript "STDP and the distribution of preferred phases in the whisker system" for consideration at PLOS Computational Biology. As with all papers reviewed by the journal, your manuscript was reviewed by members of the editorial board and by several independent reviewers. The reviewers appreciated the attention to an important topic. Based on the reviews, we are likely to accept this manuscript for publication, providing that you modify the manuscript according to the review recommendations.

Sincerely,

Abigail Morrison

Associate Editor

PLOS Computational Biology

Lyle Graham

Deputy Editor

PLOS Computational Biology

[LINK]

Reviewer's Responses to Questions

**Comments to the Authors:**

Reviewer #1: In the manuscript the authors provide a theoretical account for a puzzling experimental observation in the whisker system, i.e., the distribution of preferred phases in the cortex is not significantly sharper than that observed in the thalamus (contrary to what one would expect according to a simple "pooling" model). The non-sharpening is, according to the theory, a result of synaptic volatility induced by spike-timing dependent plasticity. The theory also makes experimental predictions that could be tested.

The results presented appear correct, as far as I can check, and novel. The assumptions underlying the modeling and its limitations are carefully discussed. The manuscript is well written and the figures informative. I found the results intriguing and certainly worth being disseminated. I have no major comment, just a curiosity: what could be the functional relevance of such a mechanism?

Reviewer #2: Neurons in the VPM and in the barrel cortex are characterized by a preferred phase. The distribution of preferred phases in the two regions is non-uniform and the widths of the two distributions is comparable. In this paper, Sherf and Shamir show that this seemingly innocuous observation is in fact highly non-trivial, and discuss a possible interpretation of this result in the framework of STDP.

I find the results of this paper both interesting and novel and I strongly recommend that it would be published in PCB. However, I think that the presentation of the results is far from optimal and be substantially improved, specifically with respect to the relationship with the biological observations.

Comments:

The important observation that the widths of the distributions are comparable is NOT supported by reference 17, as claimed in line 19. I do not doubt the observation and in fact, there is evidence for it in Yu et al., Nature Neuroscience, 2016 and Kleinfeld and Deschenes, Neuron 2011, references that are cited elsewhere in the manuscript. However, I did not find it in reference 17.

Another related point that should be clarified with respect to the relation with biology is the difference between measurements that are done simultaneously and measurements done in the same animal in different days – what can we learn from the existing literature.

In order to explain why the fact that the two distributions are comparable is surprising, the authors discuss random or uniform pooling. Explaining why such pooling is even relevant is done only in the Discussion, even this only to some extent in the framework of random synaptic changes.

I did not understand lines 244-247 and 251-252 of the Discussion section – they need to be clarified.

Reviewer #3: Rat and mice detect objects by rhythmically swiping there whiskers.

Thalamic neurons which track the phase of the whisking cycle encode

information about the location of the whisker. These neurons respond preferentially to

a particular phase which are narrowly distributed. Cortical neurons pool the rhythmic

excitation from several tens of thalamic neurons. Therefore, one may expect

a very narrow distribution of preferred phase of layer 4 cortical neuron narrow . However, this is not

observed experimentally. The paper argues that this issue can be solved if ihalamo-cortical synapses

exhibit short term dependent plasticity. To this end, the authors combine an analytical calculations

and simulations.

This is is very nice piece of work. The approach is elegant. The results are convincing.

The paper is very well written and I had pleasure to read it. I have only a couple of comments.

1) Has mentioned by the authors at the end of the discussion, they neglected the recurrent interaction

in layer 4. This should be emphasized much earlier.

Recurrent interactions - especially inhibition- may have a key role in broadening the

distribution of preferred phases in layer 4. This is because inhibition can de-correlate neuronal

activity. I suggest to the authors to elaborate a bit more about this alternative hypothesis.

2) In fact the paper determines the distribution of the preferred phase of the aggregate

thalamic input into cortical neurons. Optogenetic techniques combined with intracellular

recordings (i.e Lien and Scanziani, 2013, in the context of V1) makes it possible to test

the hypothesis of the paper. I think that commenting about that can be inspiring to the reader.

**Have the authors made all data and (if applicable) computational code underlying the findings in their manuscript fully available?**

Reviewer #1: None

Reviewer #2: Yes

Reviewer #3: Yes

PLOS authors have the option to publish the peer review history of their article (what does this mean?). If published, this will include your full peer review and any attached files.

Reviewer #1: No

Reviewer #2: No

Reviewer #3: No

Figure Files:

Data Requirements:

Reproducibility:

References:

---

## [Decision Letter · Decision Letter 1]

17 Aug 2021

Dear Mr. Sherf,

We are pleased to inform you that your manuscript 'STDP and the distribution of preferred phases in the whisker system' has been provisionally accepted for publication in PLOS Computational Biology.

Best regards,

Abigail Morrison

Associate Editor

PLOS Computational Biology

Lyle Graham

Deputy Editor

PLOS Computational Biology

Reviewer's Responses to Questions

**Comments to the Authors:**

Reviewer #2: The authors have successfully addressed all my concerns.

I have a single additional minor suggestion.

The motivation to this paper is the observation that synapses are volatile and hence are expected to converge, in the absence of activity-dependent synaptic plasticity, to random connections. By contrast, in Figure 2c, as well as throughout the text, the authors compare the experimental data to two baseline models, one in which pooling is uniform and the other in which pooling random. It is not clear to why the uniform model is biologically-relevant, as synaptic volatility is expected to result in random connectivity, not uniform.

I think that the clarity of the paper would be improved if the authors remove the uniform-model from the figure and text. However, I leave this decision to the authors.

Reviewer #3: The authors have taken into account all my comments.

**Have the authors made all data and (if applicable) computational code underlying the findings in their manuscript fully available?**

Reviewer #2: Yes

Reviewer #3: None

PLOS authors have the option to publish the peer review history of their article (what does this mean?). If published, this will include your full peer review and any attached files.

Reviewer #2: No

Reviewer #3: No

---

## [Editor Report · Acceptance letter]

3 Sep 2021

PCOMPBIOL-D-21-00842R1 

STDP and the distribution of preferred phases in the whisker system

Dear Dr Sherf,

I am pleased to inform you that your manuscript has been formally accepted for publication in PLOS Computational Biology. Your manuscript is now with our production department and you will be notified of the publication date in due course.

With kind regards,

Katalin Szabo
